# Surfactant Mediated Accelerated and Discriminatory In Vitro Drug Release Method for PLGA Nanoparticles of Poorly Water-Soluble Drug

**DOI:** 10.3390/ph15121489

**Published:** 2022-11-29

**Authors:** Ritu Gupta, Yuan Chen, Mahua Sarkar, Huan Xie

**Affiliations:** College of Pharmacy and Health Sciences, Texas Southern University, 3100 Cleburne Street, Houston, TX 77004, USA

**Keywords:** PLGA, nanoparticles (NPs), accelerated in vitro release, discriminatory, surfactant, sink conditions, AC1LPSZG, Sotax, USP apparatus 4

## Abstract

In vitro drug release testing is an important quality control tool for formulation development. However, the literature has evidence that poly-lactide-co-glycolide (PLGA)-based formulations show a slower in vitro drug release than a real in vivo drug release. Much longer in vitro drug release profiles may not be reflective of real in vivo performances and may significantly affect the timeline for a formulation development. The objective of this study was to develop a surfactant mediated accelerated in vitro drug release method for the PLGA nanoparticles (NPs) of a novel chemotherapeutic agent AC1LPSZG, a model drug with a poor solubility. The Sotax USP apparatus 4 was used to test in vitro drug release in a phosphate buffer with a pH value of 6.8. The sink conditions were improved using surfactants in the order of sodium lauryl sulfate (SLS) < Tween 80 < cetyltrimethylammonium bromide (CTAB). The dissolution efficiency (DE) and area under the dissolution curve (AUC) were increased three-fold when increasing the CTAB concentration in the phosphate buffer (pH 6.8). Similar Weibull release kinetics and good linear correlations (R^2^~0.99) indicated a good correlation between the real-time in vitro release profile in the phosphate buffer (pH 6.8) and accelerated release profiles in the optimized medium. This newly developed accelerated and discriminatory in vitro test can be used as a quality control tool to identify critical formulation and process parameters to ensure a batch-to-batch uniformity. It may also serve as a surrogate for bioequivalence studies if a predictive in vitro in vivo correlation (IVIVC) is obtained. The results of this study are limited to AC1LPSZG NPs, but a similar consideration can be extended to other PLGA-based NPs of drugs with similar properties and solubility profiles.

## 1. Introduction

PLGA is a non-toxic and biodegradable polymer with a long-term storage, better targeting abilities and tunable drug release properties [1]. Hence, PLGA-based formulations are being investigated to treat a range of diseases (e.g., diabetes, malaria, tuberculosis and cancer) [2,3,4,5] using different delivery routes (e.g., oral, transdermal, vaginal and parenteral) [6,7,8,9]. In vitro drug release testing is an important quality control tool for the development of new drug formulations including PLGA-based formulations. The choice of dissolution medium becomes crucial for the development of a robust and discriminatory in vitro drug release method. The maintenance of the sink condition is recommended, particularly for poorly water-soluble drugs, whose intestinal absorption is likely dissolution-rate limited.

The sink condition refers to the scenario when a drug’s saturation solubility, in the selected dissolution medium, is at least three-times the drug concentration [10]. In other words, the volume of the dissolution medium must be at least three to ten times of the saturation volume [11]. Sink conditions ensure that the concentration gradient is always maintained, and the drug amount already dissolved in the media is swept away and does not slow down the dissolution rate. The absence of sink conditions may lead to unpredictable release kinetics and suppressed release profiles. For realistic dissolution testing, sink conditions are preferred to achieve a fast and complete dissolution.

It has been reported in the literature that the in vitro drug release from PLGA-based formulations is much slower than the in vivo drug release [12,13]. It could be due to several in vivo factors. For example, a released drug is rapidly absorbed into the blood and transported away from the administration site. Whereas most in vitro dissolution apparatuses have closed set-ups and the drug saturation might result in a loss of sink and unreal release kinetics [14]. Further, lipids or some other biologicals may have a plasticizing effect and facilitate the water uptake into the polymer [15]. Additionally, the release of free radicals by inflammatory cells, the presence of enzymes, autocatalysis due to the accumulation of acidic degradation products, the local temperature and pH changes might result in a faster PLGA degradation in vivo [16,17,18,19,20]. Hence, an accelerated in vitro release method can potentially be a more relevant quality control tool for PLGA NPs.

Accelerated in vitro release methods with a good discriminatory ability are considered to be important for the development of an in vitro–in vivo correlation (IVIVC) [21]. Factors that can accelerate the in vitro drug release from PLGA-based formulations include the temperature, pH and the presence of enzymes or surfactants. Hu, X et al. successfully established a good IVIVC using the accelerated in vitro release method for Risperidone-loaded PLGA microspheres [22]. Susan D’Souza et al. obtained a linear Level A IVIVC between the in vitro and in vivo release profiles of the PLGA microspheres of Olanzapine using a surfactant-based release media [23]. Incecayir, T. et al. have also demonstrated that an accelerated release method using a buffer with a pH value of 6.8 and containing a nonionic surfactant was able to reflect the bioequivalence of generic tablets to the reference tablet of carvedilol [24]. In another study, an accelerated test employing a pH 6.8 phosphate buffer with a surfactant (sodium lauryl sulfate) was able to differentiate the dosage forms (tablets and capsule) with different bioavailability characteristics [25]. Hence, we developed the surfactant-mediated accelerated in vitro drug release method for PLGA NPs. To the best of our knowledge this is the first attempt to investigate a surfactants-mediated accelerated in vitro release method for PLGA NPs using Sotax USP apparatus 4 (Figure 1).

The drug release might be accelerated due to the facilitated polymer hydration/degradation or the drug diffusion [21,26,27]. Surfactants are often added in the dissolution medium to enhance the solubility of a poorly water-soluble drug by reducing the surface tension of the medium. Above the critical micelle concentration (CMC), these surfactants aggregate into micelles and solubilize a poorly water-soluble drug in their hydrophobic cores [28]. Generally, there is a linear relation between the solubility of the drug and surfactant concentration above a CMC [29]. In the present study, the effects of different surfactants on the drug solubility, sink conditions and in vitro drug release behavior have been demonstrated. Three different surfactant types, anionic (sodium lauryl sulfate or SLS), non-ionic (Tween 80) and cationic (cetyltrimethylammonium bromide or CTAB), were tested. Surfactant concentrations were chosen based on previous studies mentioned in the literature [28,30,31].

The recently identified synthetic compound AC1LPSZG, a new generation mTORC1/2 inhibitor, was used as a poorly water-soluble model compound for this work. Chemically, AC1LPSZG is (2E)-3-(4-bromophenyl)-2-(phenylsulfonyl)-*N*-(pyridine-3-ylmethyl) prop-2-enamide with a molecular formula C_21_H_17_BrN_2_O_3_S (Figure 2) and it has shown anti-NSCLC (non-small cell lung carcinoma) potential in in vitro studies [32].

Several studies support the discriminatory potential of USP 4 for in vitro drug release testing from different nano drug delivery systems. The simulation of intraluminal hydrodynamics, an excellent flow symmetry and uniformity (due to 1 mm glass beads and a red ruby bead in the conical part of the dissolution cell), a change in the media composition without disturbing the test hydrodynamics, having no requirement of an additional filtration step, a change in the pH, the temperature and the flow rate over the course of the entire test include its unique advantages over other dissolution apparatuses [33,34,35,36,37,38]. Hence, USP-4 apparatus CE7-smart (SOTAX^®^) was used to develop an accelerated and discriminatory drug release method for PLGA NPs. The developed in vitro drug release method was able to discriminate among the release profiles from the NPs prepared using a different PLGA-grade polymer [39].

## 2. Results and Discussion

AC1LPSZG-loaded PLGA NPs were prepared using the nanoprecipitation method. Three different grades of PLGA polymer, namely PLGA (50:50), PLGA (75:25) and Resomer RG 503H, were used to prepare the NPs. Their physicochemical characterization is presented in Table 1. All the particles were in a <150 nm size range with a low Polydispersity Index or PdI indicating monodisperse system. NPs prepared with Resomer RG 503 H showed the highest %EE. It could be due to the higher affinity of the ampholytic drug with free carboxyl end groups present in the polymer chain [40]. Additionally, free acidic end groups might be responsible for the more negative zeta potential of NPs prepared with Resomer RG 503H.

### 2.1. Differential Scanning Calorimetry (DSC)

DSC provides information on the thermal transformations in the NPs and the state of the drug after the encapsulation process. The DSC thermogram of pure drug AC1LPSGZ (Figure 3a) shows the endothermic peak for melting at 167.84 °C. The endothermic peak provides evidence of the crystalline nature of the drug. The absence of an AC1LPSGZ melting peak (Figure 3b) indicates that the encapsulated drug has transformed to the amorphous phase during NPs’ preparation.

### 2.2. Solubility of AC1LPSZG in Different pH Buffers

A meaningful IVIVC is expected mostly in case of BCS class II drugs where the dissolution is the rate-limiting step in the absorption [41]. As per the FDAs Guidance, the solubility of a drug compound should be tested over the pH range of 1.2–6.8 (including buffers at pH values 1.2, 4.5 and 6.8) to establish its BCS (Biopharmaceutics Classification System) class [42].

As AC1LPSZG is a novel compound, its solubility was tested in five different pH buffers at pH 1.2 (HCl buffer), pH 4.5 (acetate buffer), pH 5.5 (acetate buffer), pH 6.8 (phosphate buffer) and pH 7.4 (phosphate buffer) to establish the drug’s solubility profile (Figure 4).

The results showed the pH-dependent solubility of AC1LPSZG with a low solubility at a neutral pH and a high solubility at an acidic pH that was in agreement with the in silico predictions of the strongest pKa (base) value of 4.7 ± 0.1 and strongest pKa (acid) value of 11.0 ± 0.5.

### 2.3. In Vitro Drug Release from PLGA-AC1LPSZG-NPs at pH 1.2 and 6.8

AC1LPSZG is a novel chemotherapeutic agent in the pre-clinical development phase. It is expected to be administered by an intravenous route for a quicker onset of action. Thus, improving the drug release at about a neutral pH will be important. The scope of this study was to develop an accelerated and discriminatory in vitro drug release method reflecting the parenteral route of administration of AC1LPSZG.

We tested the in vitro drug release at pH 6.8 as well as pH 1.2 for a future investigation of an oral route of administration [43] (Figure 5). And, in that case the higher drug release in the stomach (pH 1.2) could be slowed down by the use of polymers showing a pH-dependent dissolution (e.g., poly(methacrylic acid)-polymethacrylate (PMAA–PMA)) [44]. However, in this study we focus on accelerating the drug release at pH 6.8 for a parenteral route of administration of AC1LPSZG.

### 2.4. Surfactant Mediated Accelerated AC1LPSZG Release from PLGA NPs

As AC1LPSZG solubility is high at an acidic pH (pH 1.2) (Figure 4), improving the solubility at pH 6.8 would be crucial to achieve a biorelevant solubility and improve the sink conditions. Hence, three different surfactants were explored to enhance the drug solubility and to accelerate the in vitro drug release in the phosphate buffer (pH 6.8).

To choose an appropriate surfactant for accelerating the in vitro drug release from PLGA NPs of AC1LPSZG, three different types of surfactants were tested, including anionic (Sodium lauryl sulfate, SLS), non-ionic (Polyethylene Glycol Sorbian Monooleate, Tween 80) and cationic (Hexadecyltrimethylammonium bromide, CTAB). CTAB was selected for further dissolution studies based on drug stability, solubility and a better improvement of the sink conditions.

#### 2.4.1. Effect of Surfactants on AC1LPSZG Solubility and Stability

The solubility and stability of AC1LPSZG were tested in a pH 6.8 phosphate buffer containing different concentrations of three different types (anionic, non-ionic and cationic) of surfactants. Figure 6 demonstrates the increase in the drug solubility at two different levels (0.5% and 1%) of surfactants SLS and Tween 80. Figure 7 shows a linear increase in the drug solubility at four different levels (0.5%, 1%, 1.5% and 2%) of the cationic surfactant CTAB. Figure 6 and Figure 7 show that the drug solubility was enhanced in the order of SLS < Tween 80 < CTAB.

The stability of AC1LPSZG at two different levels of SLS and Tween 80 is shown in Figure 8. The stability of the drug at four different levels of CTAB is shown in Figure 9. To check the drug stability with all three surfactants (SLS, Tween 80 and CTB), two different surfactant levels (0.5% and 1%) were tested initially. The drug was stable at all the tested levels of Tween 80 and CTAB. However, the drug was not stable with SLS. This drug instability with SLS might be explained by an interaction between the ampholytic drug and the anionic surfactant. The literature has evidence of the formation of the poorly soluble lauryl sulfate of the drug [28,45]. The presence of hydrophobic methyl groups in SLS could have dominated the drug-surfactant salt crystals’ surfaces, resulting in an increased hydrophobicity and reduced wettability by an aqueous media [46]. For this reason, SLS was found to be unsuitable for this drug. Further, the drug solubility was improved to greater extents with Tween 80 and CTAB (Figure 6 and Figure 7), but better sink conditions (explained in Section 2.4.2.) were obtained with CTAB in comparison with Tween 80. Hence, CTAB was selected as the optimal surfactant and tested at two more concentration levels (1.5% and 2%). A seemingly lower stability at a lower CTAB concentration might be due to an experimental error. A lower drug solubility at a lower CTAB concentration (Table 2) might have resulted in higher error bars.

#### 2.4.2. Effect of Surfactants on Sink Conditions

To achieve the sink conditions, the volume of the dissolution medium should be at least three to ten times of the saturation volume. In this study, the effect of different surfactants on the sink conditions and in vitro drug release has been demonstrated. SLS and Tween 80 were not able to achieve sink conditions at the tested concentrations. Relative sink conditions were improved only up to 1.3 and 2 for 1% SLS and 1% Tween 80, respectively. Whereas a much better relative sink of 8.8 was achieved with 1% CTAB. As CTAB showed a significant improvement in the sink conditions (Table 2), it was tested further at two more concentrations: 1.5% and 2% (Figure 7).

#### 2.4.3. In Vitro Drug Release with Different % CTAB in Phosphate Buffer pH 6.8

The drug release from NPs prepared using the PLGA (50:50) polymer was tested at different levels of CTAB (0.5%, 1%, 1.5% and 2%). About a three-fold increase in the DE and AUC was obtained on an increasing surfactant concentration up to 2% (Table 3). The drug release increased to 31.1% at the 2% CTAB level in comparison to only a 9.6% drug release in the phosphate buffer with a pH value of 6.8 (without surfactant) at the end of the 72 h (Figure 10).

The rate of the drug release in the phosphate buffer (pH 6.8) with a surfactant and in the phosphate buffer (pH 6.8) without a surfactant were significantly different. The dissolution efficiency (DE) is defined as the area under the dissolution curve (AUC), between the specified time points, expressed as a percentage of the area of a hypothetical rectangle describing a 100% dissolution in the same duration of time. Significant *p*-values for the DE shown in the univariate ANOVA results table followed by Tukey’s HSD (honestly significant difference) post hoc test (α = 0.05) (Table 4) showed a % drug release at all the tested levels of CTAB (0.5%, 1%, 1.5% and 2%) was significantly different from the drug release in the phosphate buffer (pH 6.8) without a surfactant (*p* < 0.05). Further, the pair-wise comparison of the DE showed that the drug release at the 2% surfactant level was significantly higher than the drug release at all other tested concentrations of CTAB (0.5%, 1% and 1.5%) (Table 4). These results demonstrate that the use of surfactant in the dissolution media has accelerated the drug release from PLGA NPs. Such an accelerated in vitro drug release test may serve as a faster quality control tool for PLGA NPs.

#### 2.4.4. Discriminatory In Vitro Drug Release Method

A discriminatory in vitro drug release method should have the capability to detect the altered drug product performance due to deliberate formulation changes. In this study, we prepared NPs using three different PLGA polymer grades and compared their drug release profiles to investigate the discriminatory potential of the developed in vitro drug release test method. The drug release from compositionally similar NPs prepared using three different PLGA polymer grades, namely PLGA (50:50), PLGA (75:25) and Resomer^®^ RG 503H, were tested at different levels of CTAB. The release profiles were more discriminated with the increase in the surfactant concentration (from 0.5% to 2%) in the dissolution medium (Figure 11).

First, the effect of an increasing CTAB concentration on the drug release was tested using PLGA (50:50) NPs with an *n* = 3 (shown in Figure 10). Table 4 shows that the drug release was accelerated on increasing the CTAB concentration from 0.5% to 2% as both the difference (DE) and the difference (AUC) values increased with the CTAB concentration. The drug release in 2% CTAB (2_CTAB) was significantly different from the drug release in a blank buffer (no_CTAB) as well as from the drug release at all other CTAB levels (i.e., 0.5%, 1% and 1.5%) (Table 4).

Subsequently, based on the above results, 2% CTAB was selected to study the drug release from the NPs prepared with two other polymers (PLGA 75:25 and Resomer RG 503H) at *n* = 3 (shown on panel (d) on Figure 11). The experiments shown in panels (a), (b) and (c) were not done in replicates (for the NPs prepared with PLGA 75:25 and Resomer RG503H) based on the results shown in Table 4, hence error bars are not added. Replicate experiments were done for all three PLGA grades only at a 2% CTAB level to confirm the discriminatory potential of the test (as shown on panel (d) on Figure 11).

One-way ANOVA followed by Tukey’s HSD post hoc test was conducted for the difference in the DE and difference in the AUC. The drug release from all three polymer grades was different from that of the drug suspension (Table 5). It is reported in the literature that the rate of the drug release is dependent on the lactide-to-glycolide ratio in the PLGA copolymer. The higher the ratio of the hydrophobic lactic acid content, the slower the drug release is due to the slower diffusion of the water molecules into the copolymer and slower hydrolysis rate [47]. This was confirmed as both the difference (DE) and the difference (AUC) were smaller for the PLGA (75:25) in comparison to the PLGA (50:50). However, the release profiles from the PLGA (50:50) and PLGA (75:25) were not discriminated in vitro probably due to the closer lactide-to-glycolide ratio. However, the drug release profile from the NPs prepared with Resomer^®^ RG 503H, (acid terminated, lactide: glycolide 50:50) was well discriminated from the drug release profiles from the NPs prepared with PLGA (50:50). Such a discrimination of the release profiles can be explained by the more hydrophilic nature of the acid-terminated Resomer^®^ RG 503H polymer. These results confirm the that developed in vitro drug release test method has the potential to discriminate the NPs formulation prepared using different polymers.

#### 2.4.5. Release Kinetics Models

Various kinetic models were fitted for the drug release from all three types of PLGA NPs. DDSolver software was used for the kinetic modelling. Figure 12 shows the visual fit of the data using different kinetic models. The Weibull model was found to best fit the dissolution data with the highest values of R2 and smallest values of AIC for all three polymers (Table 6).

The Weibull model can be represented by Equation (1). Where, F is the fraction dissolved, t is the time, Ti is the dissolution lag time, α is the time constant, β is the shape constant (slope of the best-fit line for data on a Weibull plot). The NPs prepared with PLGA (75:25) showed a slower release (highest values for Td, T50, T75 and T80) (Table 7) than the NPs prepared using PLGA (50:50). The results were in agreement with the literature, suggesting that the higher the lactide-to-glycolide ratio, the slower the degradation rate [47]. The NPs prepared using Resomer RG 503H (acid terminated, lactide: glycolide 50:50) showed the fastest release due to its hydrophilic nature (smallest values for Td, T50, T75 and T80) (Table 7).
F = 100 × {1 − Exp[−((t − Ti)^β)/α]}(1)

In the Weibull kinetic model, a β value ≤ 0.75 indicates a drug release from the NPs prepared using all three polymer types following Fickian diffusion. Comparable β values (Table 7) also suggest that the shape of the release profiles was similar.

The correlation between the “real-time” release profile in the phosphate buffer (without any surfactant) and the “accelerated” release profile (using surfactant) [48] was established using the “CORREL” in-built function in Excel (for Pearson’s correlation). A good correlation (correlation coefficient value of approximate 0.99) between the accelerated and real time releases confirms that the release mechanism was not altered in the developed accelerated test (Table 8 and Figure 13). In other words, the use of surfactant only hastens the rate of the drug release without modifying the drug release mechanism to enable logical comparisons of the real and accelerated profiles from PLGA NPs.

## 3. Materials and Methods

### 3.1. Materials

Poly (D, L-lactide-co-glycolide) lactide: glycolide 50:50 (molecular weight 30,000–60,000), Poly (D, L-lactide-co-glycolide) lactide: glycolide 75:25 (molecular weight 66,000–107,000) and Resomer^®^ RG 503H, Poly (D, L-lactide-co-glycolide), acid terminated, lactide: glycolide 50:50 (molecular weight 24,000–38,000) were all purchased from Sigma-Aldrich, St, Louis, MO, USA. AC1LPSZG was provided by our collaborator at Baylor College of Medicine. Spectra/Por^®^ Float-A-Lyzer G2 dialysis cells (cellulose ester, amber color cap, 1 ml, 300 kDa molecular weight cut–off (MWCO)) were purchased from Spectra Labs, Spectrum Laboratories Inc., Rancho Dominguez, CA, USA. Kolliphor^®^ P188 (Poloxamer 188 or Lutrol^®^ F68) and Hexadecyltrimethylammonium bromide (or Cetrimonium Bromide, CTAB) and was purchased from Sigma-Aldrich, St, Louis, MO, Tween^®^ 80 (Polyoxyethylene (80) Sorbitan Monooleate) was purchased from EMD Chemicals Inc.,Gibbstown, NJ, USA) and Sodium lauryl sulfate was bought from Bio-Rad Laboratories, Hercules, CA, USA.

### 3.2. Preparation of PLGA-AC1LPSZG-NPs

The PLGA NPs of AC1LPSZG were prepared using a solvent displacement/nanoprecipitation method. Briefly, 60 mg of PLGA were dissolved in 2 mL of acetone. A total of 5 mg of AC1LPSZG were dissolved in the above prepared polymer solution and then added dropwise into the aqueous phase containing a stabilizer (3% Poloxamer P188) under magnetic stirring (0.3 mL/min, 30 °C, 750 rpm) and using a syringe pump (Model: NE-300) (New Era Pump Systems, Inc., Farmingdale, NY, USA) attached with a Heidolph magnetic stirrer (Heidolph Instruments Gmbh & co., Schwabach, BY, Germany). A solvent removal was done under magnetic stirring on hot plate stirrers (NO97042-634) (VWR, Troemner LLC, USA) at 60 °C, 400 rpm for 3 h. The NPs were washed three times with water at 14,000 rpm, 4 °C for 45 min using Eppendorf centrifuge 5417R (Rotor no. F45-30-11) from CE, Germany and lyophilized (SP Scientific BenchTop Pro with Omnitronics^TM^) using 10% sucrose as lyoprotectant.

### 3.3. Particle Size & Zeta Potential Measurement

Malvern Zetasizer Nano-ZS (Model: ZEN 3600) (Malvern Instrument Ltd., Worcestershire, UK) was used for the particle size (Appendix, Figure A1) and zeta potential (Appendix, Figure A2) measurements. Disposable polystyrene folded capillary cells (DTS0012, Malvern, UK) were rinsed with dispersant before use. Milli Q water was used as a dispersion medium during all of the measurements.

### 3.4. Entrapment Efficiency (%EE)

For the entrapment efficiency determination, 20 mg NPs were sonicated with 1.5 mL of acetonitrile for 5 min (Bath sonicator, B2500A-MTH, VWR International, West Chester, PA, USA) to dissolve the PLGA and then centrifuged at 11,000 rpm, 4 °C for 10 min in a cooling centrifuge (Eppendorf centrifuge 5417R, Rotor no. F45-30-11, CE, Eppendorf, Germany) to precipitate out the sucrose lyoprotectant. The supernatant was collected and analyzed under a Waters Acquity UPLC system after a suitable dilution and centrifuging again at 11,000 rpm, 4 °C for 10 min.
(2)$$\%\;\mathrm{EE}=\frac{\mathrm{Amount}\;\mathrm{of}\;\mathrm{drug}\;\mathrm{encapsulated}\;}{\mathrm{Amount}\;\mathrm{of}\;\mathrm{drug}\;\mathrm{initially}\;\mathrm{taken}\;\mathrm{to}\;\mathrm{prepare}\;\mathrm{NPs}}\times 100$$

### 3.5. In Vitro Drug Release Study

The in vitro drug release of AC1LPSZG-loaded PLGA NPs was tested in a phosphate buffer with a pH value of 6.8 for 72 h. Three different surfactants were explored to enhance the drug solubility and accelerate the in vitro drug release. Adding the surfactants did not change the pH value of the buffer. A close loop type USP-4 apparatus CE7-smart (SOTAX^®,^ Hamburg, Germany) with 22.6 mm dissolution cells (14 mL) and a piston pump (Sotax™CP7-35/CP7-300, Hamburg, Germany), incorporated with Float-A-Lyzer dialysis cells at a 300 kDa molecular weight cut-off (MWCO) was used for the in vitro drug release studies (Figure 2).

A known amount of drug-loaded PLGA NPs was suspended in 1 mL of respective dissolution medium and filled in Float-A-Lyzer dialysis cells. In total, 100 mL of the dissolution medium was filled in each media bottle. The test method was loaded manually; the sample volume, flow rate of the release medium and temperature were set at 200 μL, 16 mL/min and 37 °C, respectively. At predetermined time intervals, the samples were withdrawn and assayed using UPLC after a suitable dilution. The experiments were done in triplicate and the data were presented as the mean ± SD.

### 3.6. Sample Preparation for LC-MS/MS Analysis by Liquid–Liquid Extraction

As very low dissolution concentrations were obtained (approximately 10% drug was released at 72 h) in a pH 6.8 buffer without a surfactant (Figure 5), a previously developed LC-MS/MS method [49] was used to analyze the drug concentrations after performing a liquid–liquid extraction procedure. The drug was extracted from the aqueous dissolution medium (phosphate buffer, pH 6.8) into an equal volume of volatile organic solvent (ethyl acetate) (Figure 14) before the LC-MS/MS analysis.

### 3.7. Differential Scanning Calorimetry (DSC)

In a heat flux-type DSC (Shimadzu DSC-60A), an empty reference pan and a sample loaded pan both were kept on a thermoelectric disk. The disk was heated at a linear heating rate and the generated temperature difference (due to heat capacity of the sample) between the sample and the reference pan was measured by area thermocouples and converted to the heat flown. Thermograms were recorded at a 10 °C per minute heat flow under the N_2_ atmosphere.

### 3.8. Analytical Methods

#### 3.8.1. Ultra-Performance Liquid Chromatography (UPLC) Method

The UPLC method was used for the in vitro sample analysis (dissolution studies and entrapment efficiency determination) of AC1LPSZG-loaded PLGA nanoparticles (PLGA-AC1LPSZG-NPs). Griseofulvin was selected as the internal standard (IS). Waters, ACQUITY UPLC BEH C18 (50 mm × 2.1 mm i.d., 1.7 µm, 100 Å) column was used with PDA detector (285 nm). The retention times for AC1LPSZG and IS were 2.20 and 2.63 min, respectively (Appendix, Figure A3). The mobile phase A was 0.1% formic acid in water, and the mobile phase B was 0.1% formic acid in acetonitrile at a flow rate of 0.5 mL/min.

#### 3.8.2. LC-MS/MS Method

For the MS/MS analysis, a previously validated method published by our lab was used [49]. Briefly, a 4000 QTRAP^®^ triple quadrupole mass spectrometer with a Turbo Ion Spray source (AB Sciex, Redwood City, CA, USA) was used in the positive mode. The MRM transitions for the [M + H]+ ion of AC1LSPZG was (m/z 457.10 → 349.00) and that of Griseofulvin (IS) was (m/z 353.27 → 285.10). Representative LC chromatogram for tandem mass spectrometry is shown in Appendix (Figure A4).

### 3.9. Statistical Data Analysis 

A one-way ANOVA test followed by Tukey’s HSD (honestly significant difference) post hoc test was used to evaluate the significant differences between the release profiles using JMP software. DDSolver software (add-in to Microsoft Excel) was used for the modelling of the release kinetic.

## 4. Conclusions

In vitro drug release testing is a key quality control tool for formulation development. However, for PLGA-based formulations, very slow in vitro drug release profiles do not reflect real in vivo performances, and therefore lead to a significant delay in the formulation development. In the literature, there is evidence where surfactant-mediated accelerated drug release methods were able to differentiate between the formulations that were not bioequivalent. Hence, we developed the surfactant-mediated accelerated in vitro drug release method for PLGA NPs. To the best of our knowledge, this is the first attempt to investigate a surfactants-mediated accelerated in vitro release method for PLGA NPs using Sotax USP apparatus 4.

In this study, the surfactant mediated accelerated and discriminatory in vitro drug release method was successfully developed for the PLGA-based NPs of the poorly soluble model compound AC1LPSZG. The study highlights the utility of surfactants for an accelerated in vitro drug release testing of the PLGA-based NPs. The release profiles obtained from the accelerated test correlated well with the “real-time” release profiles (obtained with no surfactant in the dissolution medium). It suggests that the release kinetics were not changed with the accelerated drug release method. These results indicate that the surfactant-based accelerated USP apparatus 4 method could serve as a fast quality control tool for PLGA-based systems. Furthermore, the accelerated in vitro release test was also able to discriminate the release profiles of the drug from NPs prepared from different PLGA grades. Our lab is currently conducting in vivo studies to further attempt an IVIVC of PLGA-based NPs. Although the results of this experiment are specific to AC1LPSZG NPs, similar considerations might be used to develop an accelerated and discriminatory in vitro release method for PLGA-based NPs of other drugs with similar properties and solubility profiles.

## Figures and Tables

**Figure 1 pharmaceuticals-15-01489-f001:**
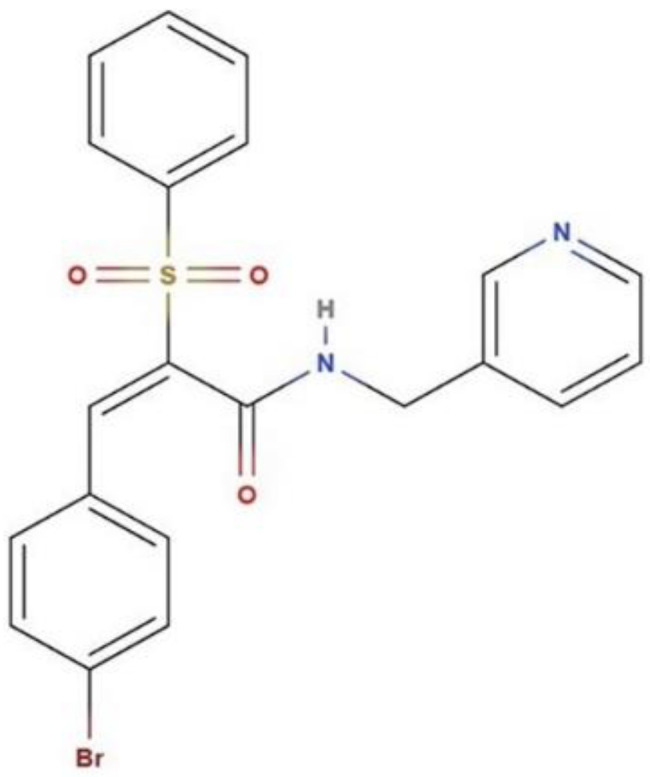
Chemical structure of compound AC1LPSZG.

**Figure 2 pharmaceuticals-15-01489-f002:**
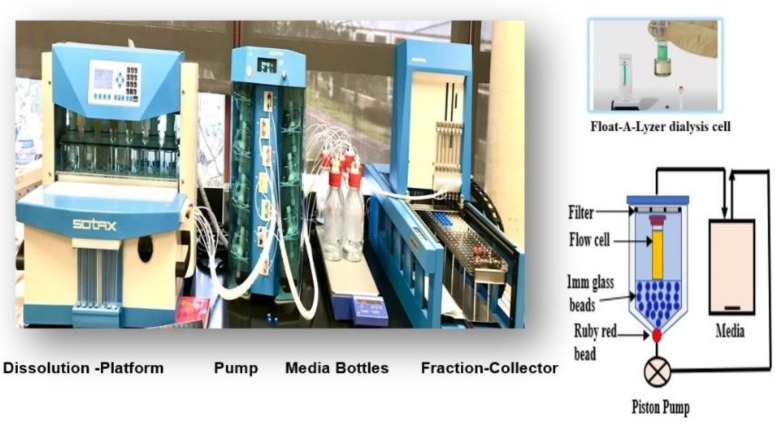
Sotax USP apparatus 4 (Closed Loop System).

**Figure 3 pharmaceuticals-15-01489-f003:**
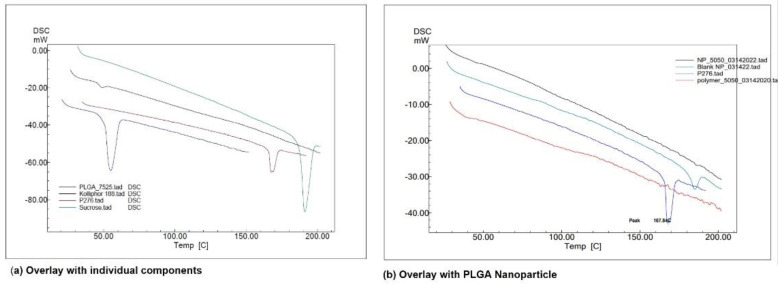
DSC thermogram using Shimadzu DSC-60A at 10 °C per min and N_2_ atmosphere. (**a**) Overlay with individual components, (**b**) overlay with PLGA NPs. (Kolliphor 188 represents stabilizer Poloxamer P188, P276 represents drug AC1LPSGZ, NP-5050 represents NPs prepared with polymer PLGA (50:50), blank NP represents empty NPs without drug, Polymer-5050 represents polymer PLGA (50:50)).

**Figure 4 pharmaceuticals-15-01489-f004:**
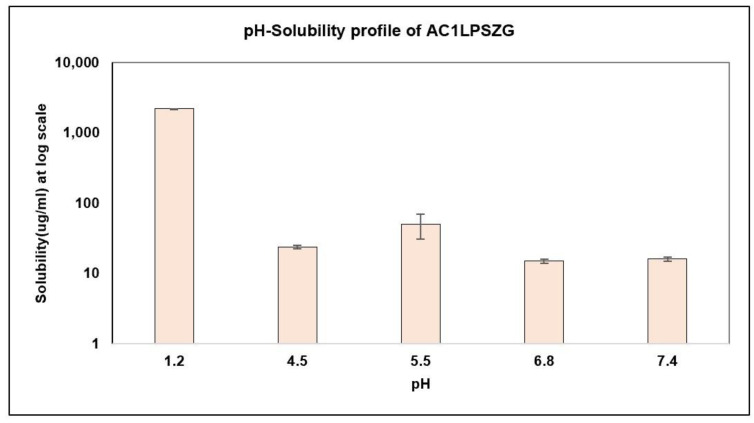
Drug Solubility in Different pH Buffers, (*n* = 3).

**Figure 5 pharmaceuticals-15-01489-f005:**
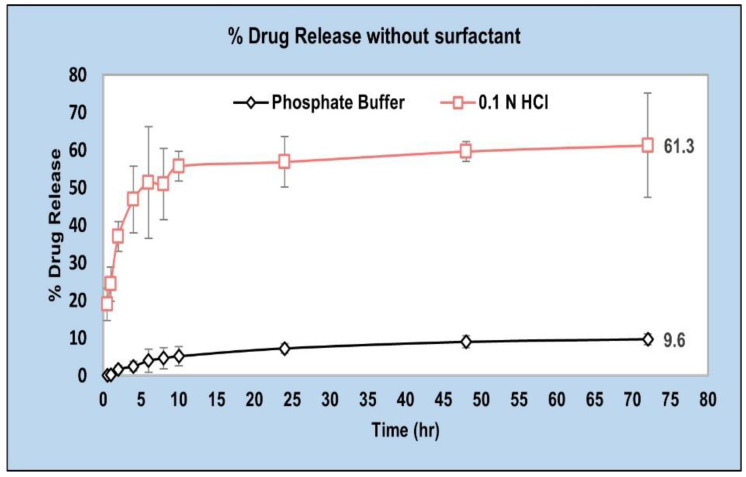
In vitro drug release from PLGA (50:50) in pH 1.2 and pH 6.8 buffers without any surfactant (*n* = 3).

**Figure 6 pharmaceuticals-15-01489-f006:**
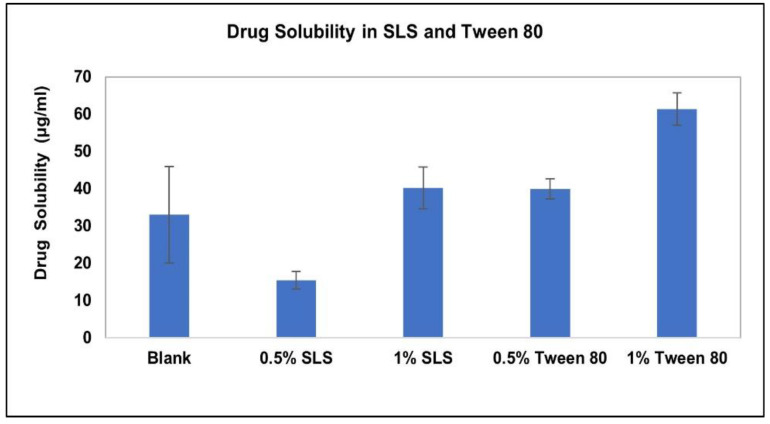
Effect of SLS and Tween 80 on AC1LPSZG solubility, (*n* = 3).

**Figure 7 pharmaceuticals-15-01489-f007:**
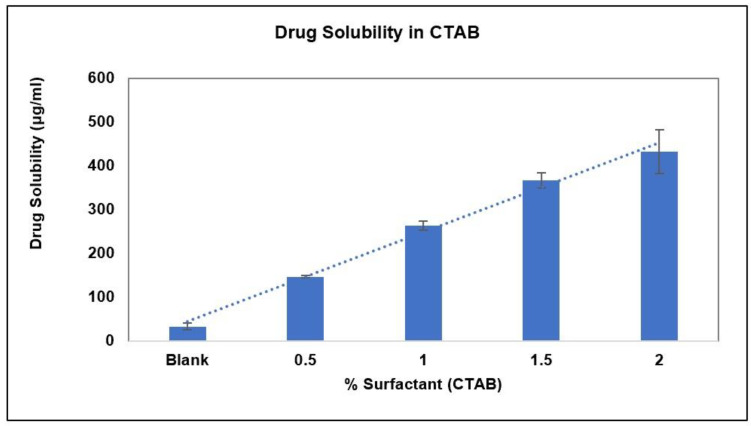
Effect of CTAB on AC1LPSZG solubility, (*n* = 3).

**Figure 8 pharmaceuticals-15-01489-f008:**
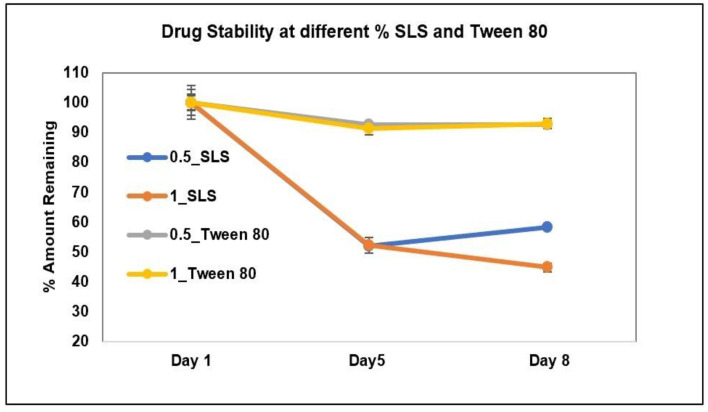
AC1LPSZG stability in pH 6.8 phosphate buffer containing two different concentrations (0.5% and 1%) of SLS and Tween 80. 0.5_SLS; 0.5% SLS, 1_SLS; 1% SLS, 0.5_Tween 80; 0.5% Tween 80, 1_Tween 80; 1% Tween 80, (*n* = 3).

**Figure 9 pharmaceuticals-15-01489-f009:**
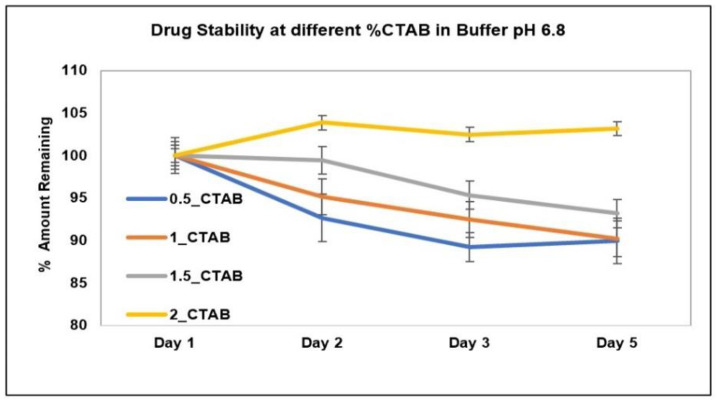
AC1LPSZG stability in pH 6.8 phosphate buffer containing different concentrations of CTAB (0.5%, 1%, 1.5% and 2%). 0.5_CTAB; 0.5% CTAB, 1_CTAB; 1% CTAB, 1.5_CTAB; 1.5% CTAB, 2_CTAB; 2% CTAB, (*n* = 3).

**Figure 10 pharmaceuticals-15-01489-f010:**
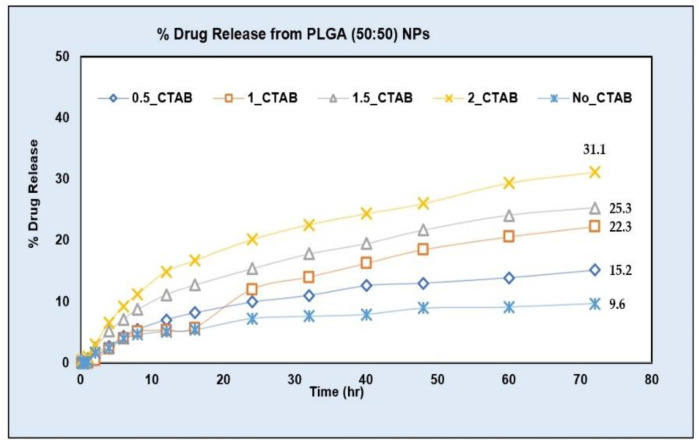
AC1LPSZG release from PLGA (50:50) NPs in phosphate buffer pH 6.8 with different concentrations of CTAB. no_CTAB; no surfactant, 0.5_CTAB; 0.5% CTAB, 1_CTAB; 1% CTAB, 1.5_CTAB; 1.5% CTAB, 2_CTAB; 2% CTAB. (*n* = 3).

**Figure 11 pharmaceuticals-15-01489-f011:**
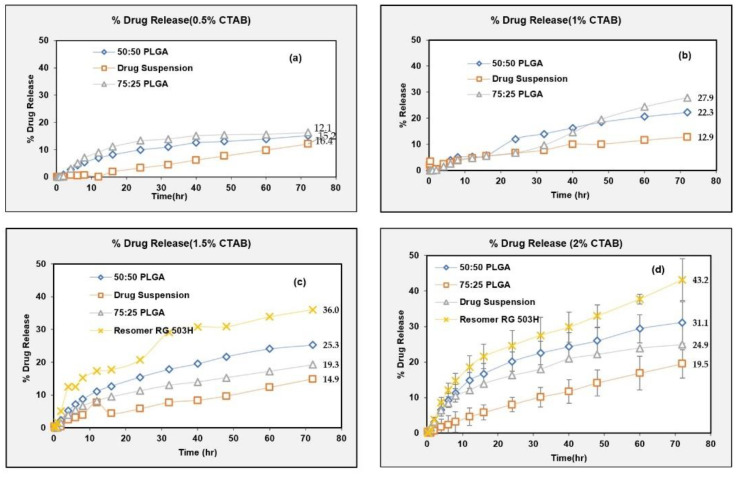
Drug release from NPs of different polymer grade at different % CTAB. (**a**) 0.5%, (**b**) 1%, (**c**) 1.5%, (**d**) 2%.

**Figure 12 pharmaceuticals-15-01489-f012:**
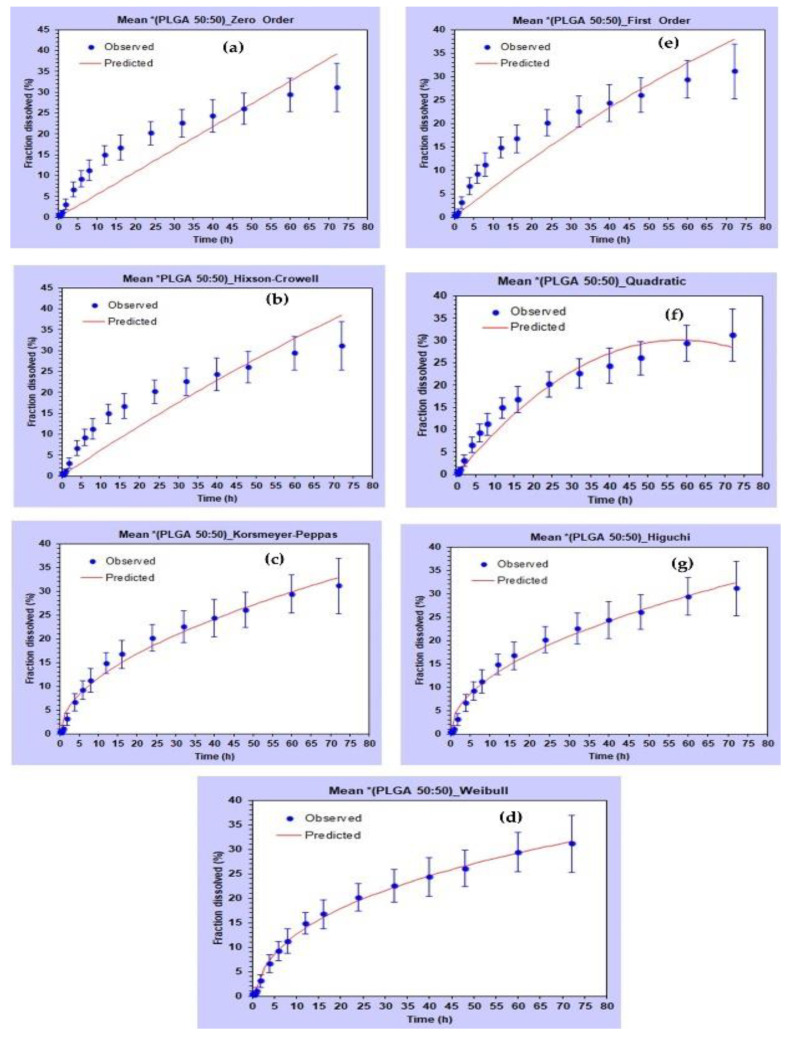
Fitting of different kinetic models for in vitro release profile of AC1LPSZG from PLGA (50:50) NPs in 2% CTAB. (**a**) Zero order, (**b**) Hixon Crowell, (**c**) Korshmeyer–Peppas, (**d**) Weibull, (**e**) First order, (**f**) Quadratic, (**g**) Higuchi.

**Figure 13 pharmaceuticals-15-01489-f013:**
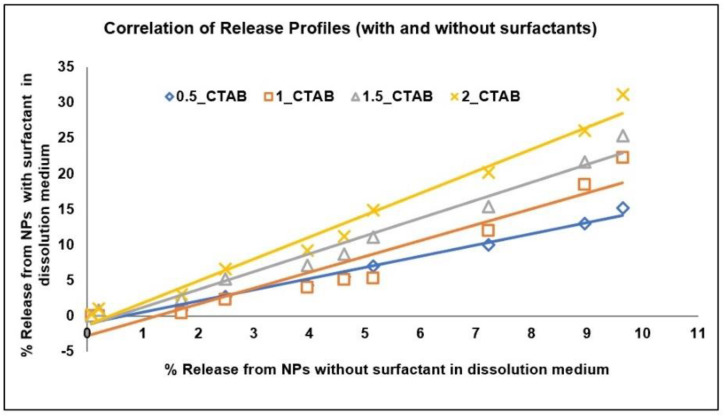
Correlation of Release Profiles (with and without surfactants). 0.5_CTAB; 0.5% CTAB, 1_CTAB; 1% CTAB, 1.5_CTAB; 1.5% CTAB, 2_CTAB; 2% CTAB.

**Figure 14 pharmaceuticals-15-01489-f014:**
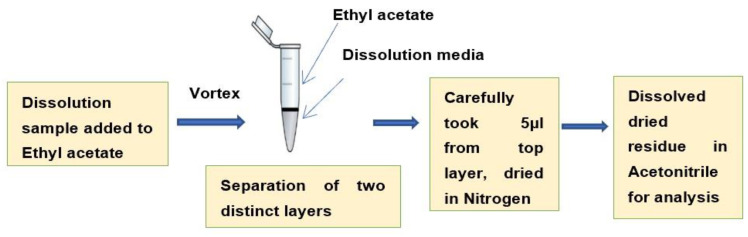
Sample preparation for LC-MS/MS analysis by liquid–liquid extraction.

**Table 1 pharmaceuticals-15-01489-t001:** Physicochemical characterization of NPs prepared using 3 different PLGA grades.

PLGA Grade	Size	PdI	Zeta	EE%
PLGA (50:50)	124 ± 6.19	0.078 ± 0.02	−15.0 ± 3.28	41.2 ± 13.30
PLGA (75:25)	133 ± 9.12	0.095 ± 0.01	−12.8 ± 3.54	38.3 ± 2.52
Resomer RG 503H	136 ± 13.21	0.060 ± 0.02	−53.6 ± 2.11	49 ± 7.55

**Table 2 pharmaceuticals-15-01489-t002:** Relative sink * conditions with different surfactant concentrations in buffer pH 6.8.

Surfactant	Saturation Solubility (Cs) (µg/mL)	Rel. Sink (Cs/Cd)
Water	24	0.8
Blank buffer (pH 6.8)	33	1.1
0.5% SLS	15	0.5
1% SLS	40	1.3
0.5% Tween 80	40	1.3
1% Tween 80	61	2.0
0.5% CTAB	147	4.9
1% CTAB	264	8.8
1.5% CTAB	366	12.2
2% CTAB	432	14.4

* Relative sink is calculated as the ratio of Cs and Cd. Where Cs is saturation solubility of drug and Cd is drug concentration after complete dissolution of NPs in 100 mL dissolution medium.

**Table 3 pharmaceuticals-15-01489-t003:** Mean dissolution efficiency (DE) and area under the dissolution curve (AUC) for PLGA (50:50) NPs at different % CTAB.

% CTAB	No_CTAB	0.5_CTAB	1_CTAB	1.5_CTAB	2_CTAB
DE (%)	0.07	0.12	0.14	0.15	0.20
AUC (percent–hour)	533	843	981	1084	1432

AUC; area under the dissolution curve, DE; area under the dissolution curve between specified time points, expressed as a percentage of area of a hypothetical rectangle describing 100% dissolution in same duration of time. no_CTAB; no surfactant, 0.5_CTAB; 0.5% CTAB, 1_CTAB; 1% CTAB, 1.5_CTAB; 1.5% CTAB, 2_CTAB; 2% CTAB. (*n* = 3).

**Table 4 pharmaceuticals-15-01489-t004:** 95% confidence intervals for the difference in DE (%) and AUC (percent–hour) between two surfactant pairs obtained from univariate ANOVA test followed by Tukey’s HSD (honestly significant difference) post hoc test (α = 0.05), (*n* = 3). DE; dissolution efficiency, AUC; area under dissolution curve, 0_CTAB; no surfactant, 0.5_CTAB; 0.5% CTAB, 1_CTAB; 1% CTAB, 1.5_CTAB; 1.5% CTAB, 2_CTAB; 2% CTAB. * *p* < 0.05.

Surfactant Pair		Difference (DE)	Tukey’s HSD Significance	Difference (AUC)	Tukey’s HSD Significance
no_CTAB	0.5_CTAB	0.043 (0.006, 0.081)	No	310 (−13, 633)	No
no_CTAB	1_CTAB	0.063 * (0.025, 0.101)	Yes	449 * (126, 772)	Yes
no_CTAB	1.5_CTAB	0.077 * (0.039, 0.115)	Yes	551 * (228, 874)	Yes
no_CTAB	2_CTAB	0.126 * (0.089, 0.164)	Yes	899 * (576, 1222)	Yes
0.5_CTAB	1_CTAB	0.020 (−0.022, 0.061)	No	139 (−215, 492)	No
0.5_CTAB	1.5_CTAB	0.034 (−0.008, 0.075)	No	241 (−113, 595)	No
1_CTAB	1.5_CTAB	0.014 (−0.027, 0.055)	No	103 (−251, 456)	No
2_CTAB	0.5_CTAB	0.083 * (0.041, 0.124)	Yes	589 * (235, 943)	Yes
2_CTAB	1_CTAB	0.064 * (0.022, 0.105)	Yes	451 * (97, 804)	Yes
2_CTAB	1.5_CTAB	0.050 * (0.008, 0.091)	Yes	348 (−6, 702)	No

**Table 5 pharmaceuticals-15-01489-t005:** 95% Confidence intervals for the difference in DE (%) and AUC (percent–hour) between different polymer pairs (at 2% CTAB) and univariate ANOVA followed by Tukey’s HSD (honestly significant difference) post hoc test (α = 0.05), DE; dissolution efficiency, AUC; area under dissolution curve.

Polymer Pair		Difference (DE)	Tukey’s HSD Significance	Difference (AUC)	Tukey’s HSD Significance
Resomer RG 503H	Drug Suspension	0.186 (0.126, 0.247)	Yes	1341 (915, 1767)	Yes
PLGA (50:50)	Drug Suspension	0.126 (0.066, 0.187)	Yes	914 (487, 1340)	Yes
PLGA (75:25)	Drug Suspension	0.086 (0.026, 0.147)	Yes	647 (220, 1073)	Yes
Resomer RG 503H	PLGA (50:50)	0.06 (−0.001, 0.121)	No	428 (2, 854)	No
PLGA (50:50)	PLGA (75:25)	0.04 (−0.021, 0.101)	No	267 (−159, 693)	No
Resomer RG 503H	PLGA (75:25)	0.1 (0.039, 0.161)	Yes	695 (268, 1121)	Yes

**Table 6 pharmaceuticals-15-01489-t006:** Kinetic models for NPs prepared using different PLGA grades. R^2^_adj.; adjusted coefficient of determination, AIC; Akaike information criterion, MSE; mean squared error, MSC; model selection criteria.

Parameter/Model	PLGA (50:50) NPs				PLGA (75:25) NPs				Resomer RG 503H NPs			
	R^2^_adj	AIC	MSE	MSC	R^2^_adj	AIC	MSE	MSC	R^2^_adj	AIC	MSE	MSC
Zero-order	0.76	97.31	27.95	1.28	0.71	94.12	22.6	1.03	0.8	103.3	42.85	1.4
First-order	0.83	91.44	20.03	1.65	0.77	90.2	17.77	1.28	0.87	97.22	28.13	1.78
Hixson-Crowell	0.81	93.49	22.47	3.49	0.75	91.54	19.28	1.2	0.85	99.26	32.5	1.65
Quadratic	0.95	73.39	6.32	2.78	0.92	70.41	6.08	2.52	0.94	86.72	12.87	2.44
Higuchi	0.97	65.3	3.8	3.28	0.97	58.33	2.41	3.27	0.96	79.27	9.03	2.9
Korsmeyer–Peppas	0.98	62	2.93	3.49	0.97	56.79	2.12	3.37	0.97	77.8	7.23	2.99
Weibull	0.99	35.22	0.59	5.16	0.99	42.84	0.79	4.24	0.98	54	3.96	4.48

**Table 7 pharmaceuticals-15-01489-t007:** Comparison of various Weibull Model parameters. Td; time for 63.2% drug release, T50; time for 50% drug release, T75; time for 75% drug release, T80; time for 80% drug release, α and β are the time and shape constant and Td is the time parameter.

Polymer Type/Parameter	Resomer RG 503H	PLGA (50:50)	PLGA (75:25)
α	24.9	22.2	23.6
β	0.56	0.49	0.45
T_d_	10.6 days	25.9 days	81.4 days
T_50_	5.1 days	11.8 days	30.2 days
T_75_	20.8 days	52.5 days	201.0 days
T_80_	28.3 days	72.7 days	305.5 days

**Table 8 pharmaceuticals-15-01489-t008:** Correlation of release profiles (with and without surfactants).

	0.5_CTAB	1_CTAB	1.5_CTAB	2_CTAB
Correlation Coefficient (R)	0.991068	0.954761	0.987057	0.990405

## Data Availability

Data is contained within the article.

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
