# Peer review of "Surfactant Mediated Accelerated and Discriminatory In Vitro Drug Release Method for PLGA Nanoparticles of Poorly Water-Soluble Drug"

_pharmaceuticals, 2022, doi:10.3390/ph15121489_

Round 1

Reviewer 1 Report

The manuscript described the development of in vitro drug release assay of PLGA nanoparticles. Having a reliable in vitro drug release assay is important in developing novel PLGA-based drug formulations, but the current assays did not reflect well in vivo. The study investigated the addition of three different surfactants (SLS, Tween 80, and CTAB) into the dissolution medium on drug solubility, sink conditions, and drug release with the use of a hydrophobic drug, AC1LPSGZ. The manuscript is mostly well-presented but still unsuitable for publication as there are some flaws and ambiguities that will need to be clarified.  

1.     Please make sure the correct statistical analysis is used. The authors seem to use Student’s t-test as the post hoc test for one-way ANOVA. However, this will cause inflation in Type I error if no adjustment is made based on the number of comparisons. Student’s t should only be used to compare two groups and not for post hoc tests after ANOVA.

2.     The authors will need to provide more discussion in the results session to reassure the readers that the experimental designs are logical and scientifically meaningful. For example, in Line 171-173, the authors stated “As AC1LPSZG solubility is high at acidic pH (pH 1.2) (Figure 4), improving the solubility at pH 6.8 would be crucial to achieve biorelevant solubility and improve the sink conditions.” However, if the authors intend to administer the drug from the oral route, the PLGA NP will still go through the stomach (pH1.2) before reaching the intestine (pH 6.8). Would it be more important to reduce the drug release during the gastric condition first before worrying about the drug solubility in the intestine? From figure 5, drug-releasing at pH 1.2 was at least 6 times higher than at pH 6.8. Will adding the surfactants also further enhance the drug release at pH1.2?

3.     The authors claim that the improved in vitro assay could reflect in vivo drug-releasing profile, which is the whole point of this study. The authors need to either provide relevant literature to support those similar modifications of the in vitro drug release assay (even for other formulations) have been shown to reflect well for in vivo/clinical conditions or demonstrate the in vitro drug release profile here is similar to the profile from in vivo experiment.   

Introduction

The last paragraph seems to be more suitable in Materials and Methods.

Results and Discussion

Figure 3: more details in describing the graph legend are needed. What is the condition for Kolliphor® P 188, polymer 5050, etc?

2.2 Solubility…: the authors should discuss why the pH other than 1.2 and 6.8 were chosen (how they are related to the potential drug administration route). Also, which buffer was used for each pH condition?

Table 2: Please indicate the pH for the blank buffer (not 6.8?). Does adding the surfactants change the pH?

Line 355-359: what statistical analysis was used to study the correlation?

Author Response

Please see attached response.

Reviewer 2 Report

Dear Authors,

  "Surfactant Mediated Accelerated and Discriminatory In Vitro Drug Release Method for PLGA Nanoparticles of Poorly Water- Soluble Drug"

The comments about the manuscript are in the following order:

1.     Line 42: In the case of the sink, referring to the drug concentration is more common than referring to the volume of the medium.

2.     What is the grade of PLGA in figure 5?

3.     In figure 8, Please provide detailed medium information in captions.

4.     In figure 9, Please provide detailed medium information in captions.

5.     In figure 9 the effect of a higher concentration of CTAB on AC1LPSZG stability is less than the lower concentration of CTAB. How It can be explained?

6.     In table 2, Please mention the Cs of AC1LPSZG.

7.     In Figure 11 SD was missed. In the curved points (a, b, and c) it should be noted.

8.     In table 3, please describe “DE” and “AUC” in the caption of the table.

10.  The role of drug structure in the release is not considered. While the release of different drugs from the same polymer base can be very different. How do you include this variable in the experiment?

11.  Lin 23-27; In “Abstract”, This study does not cover this major outcome. It is better to use inference only about PLGA NPs.

12.  Lin 483-485; In “Conclusions”, This conclusion can be used for PLGA NPs only and it Can’t expand further in this experiment.

Regards,

Author Response

Please see attached Response. 

Round 2

Reviewer 1 Report

The authors addressed all my comments. The reference should be checked again to ensure the correct literature is cited for the updated content. 

Author Response

Dear Reviewer,

We sincerely appreciate your time to review our manuscript and provide valuable comments. We have revised the manuscript as per your suggestions and hope the corrections meet with journal approval requirements.

Thank You!

Best Regards,  

Comment 1: The authors addressed all my comments. 

Author’s Response: Thank you again for valuable comments.

Comment 2:  The reference should be checked again to ensure the correct literature is cited for the updated content. 

Author’s Response:  References are updated to match the updated content. 

Reviewer 2 Report

Dear Authors,

 -        In a part of the discussion, according to the reviewer's opinion, there is the extension of the result beyond the test, which is suggested to be removed.) It is highlighted (

Line 544: "In brief, similar discriminatory test methods can be used as a quality control tool to identify critical formulation and process parameters and can also be used as a surrogate for bioequivalence studies if a predictive IVIVC (In vitro In vivo correlation) is obtained.”

-         In figure 12, (a) was missed. 

Regards,

Author Response

Dear Reviewer,

We sincerely appreciate your time to review our manuscript and provide valuable comments. We have revised the manuscript as per your suggestions and hope the corrections meet with journal approval requirements.

Thank You!

Best Regards,  

Comment 1: 

In a part of the discussion, according to the reviewer's opinion, there is the extension of the result beyond the test, which is suggested to be removed.) It is highlighted (

Line 544: "In brief, similar discriminatory test methods can be used as a quality control tool to identify critical formulation and process parameters and can also be used as a surrogate for bioequivalence studies if a predictive IVIVC (In vitro In vivo correlation) is obtained.”

Author’s Response: Line 544 was deleted as suggested.

Comment 2:  In figure 12, (a) was missed. 

Author’s Response: In figure 12, (a) was added in the figure caption.